# Effect of Silica Fume on Engineering Performance and Life Cycle Impact of Jute-Fibre-Reinforced Concrete

Rawaz Kurda [1,2,3] 

1   Department of Highway and Bridge Engineering, Technical Engineering College, Erbil Polytechnic University, Erbil 44001, Iraq; rawaz.kurda@tecnico.ulisboa.pt
2   Department of Civil Engineering, College of Engineering, Nawroz University, Duhok 42001, Iraq
3   CERIS, Civil Engineering, Architecture and Georresources Department, Instituto Superior Técnico, Universidade de Lisboa, Av. Rovisco Pais, 1049-001 Lisbon, Portugal

**Abstract:** The brittleness of plain concrete (PC) is a result of its lack of tensile strength and poor resistance to cracking, which in turn limits its potential uses. The addition of dispersed fibres into the binding material has been demonstrated to have a positive impact on the tensile properties of PC. Nevertheless, using new or engineered fibres in concrete significantly increases the overall cost and carbon footprint of concrete. Consequently, the main obstacle in creating environmentally friendly fibre-reinforced concrete is the traditional design process with energy-intensive materials. This study investigated how the engineering properties and life cycle impact of concrete were influenced by varying the volume fractions of jute fibre (JF). The impact of incorporating silica fume (SF) as a partial replacement of Portland cement was also studied. The studied parameters included mechanical behaviour, non-destructive durability indicators, and the life cycle impact of concrete using JF and SF. The efficiency of JF in mechanical performance improved with the increase in age and with the addition of SF. When using both SF and 0.3% JF, there was an improvement of around 28% in the compressive strength (CS). When 0.3% JF was added, in the presence and absence of SF, the splitting tensile strength (STS) improvement was around 20% and 40%, respectively. The addition of JF improved the residual flexural strength (FS) and flexural ductility of PC. The SF addition overcame the drawbacks of the poor resistance of JF-reinforced concrete (JFRC) against water absorption (WA) and rapid chloride ion penetration (RCIP).

**Keywords:** cellulose fibre; cement replacement; fibre reinforcement; flexural ductility; life cycle; engineering properties



## 1. Introduction

High-performance concrete (HPC) is created with the specific purpose of being more resilient and having greater strength compared to conventional PC systems. HPC achieves its high CS by utilizing a low water-cement (w/c) ratio, high cementitious material content, low aggregate volume, and "supplementary cementitious materials" (SCMs), resulting in a denser microstructure [1]. It is commonly understood that PC has a quasi-brittle nature, meaning that its brittleness increases as its strength increases [2]. Consequently, the primary concerns when using high-strength grades of PC are the increased brittleness and reduced ability to withstand cracking and spreading [3]. Cracking in PC can occur from various causes, including tensile loads and micro-cracks resulting from plastic and/or autogenous shrinkage during pre-hardening or drying shrinkage in the hardened state [4,5]. Inherent cracks allow harmful fluids to penetrate, compromising the durability of the concrete structure by subjecting it to detrimental chemicals [6,7]. Improving the tensile performance, crack resistance, and strain-hardening response of PC is a significant area of focus in the field of concrete technology [8].

Incorporating dispersed reinforcement/fibres into the PC matrix can address the problem of brittleness, resulting in a material with enhanced tensile strength, impact

resistance, and energy/shock absorption capacity [9–12]. The impact of fibres on concrete's properties depends on multiple factors, including the material properties of the fibres, the size and shape of the fibres, the concentration and orientation of the dispersed fibres within the PC matrix, and the interfacial properties of the fibre–matrix interaction [13–16]. The impact of fibres on the characteristics of concrete is influenced by various factors, such as the fibre's material properties, as well as its size and shape [17,18]. Typically, engineered fibres, which are new or unused, are traditionally produced from materials such as glass, steel, polymers, and carbon fibres [19,20]. However, the manufacture and processing of these fibres require significant amounts of energy, resulting in substantial greenhouse gas emissions. There is a growing interest in evaluating natural plant-based fibres as an alternative to engineered fibres in concrete, given the environmental concerns associated with the manufacturing and processing of the latter [12,21,22]. Plant-based fibres are obtained from renewable, inexpensive, and recyclable sources, i.e., plants. Various kinds of plant fibres, such as sisal, coir, jute, wheat straws, hemp, etc., have been investigated as potential fibre reinforcements in concrete [12,23–25].

Jute fibre (JF) is regarded as an affordable and extremely durable natural fibre, among other plant-based fibres. Approximately 81% of the world's overall jute production is attributed to India and Bangladesh [26]. JF typically comprises plant components such as "cellulose" and "lignin". It possesses various beneficial properties, such as axial tensile strength, modulus of resilience, re-useability, a long life, and bio-degradation, among others. These properties make it a more advantageous option than other plant-based fibres [22,27].

Numerous studies have been conducted thus far to explore the characteristics of concrete reinforced with JF. Zakariya et al. [28,29] conducted a study on how different percentages and lengths of JF affect the properties of normal concrete. According to the research, the inclusion of JF at a smaller volume percentage led to an improvement in the strength parameters of the concrete, including CS, FS, and STS. The study also found that the optimal efficiency of JF occurred at fibre lengths of 10–15 mm. Ozawa and colleagues [30] discovered that the addition of 0.075% JF to HPC can rectify its brittle behaviour and serve as a protective measure against fire spalling. Sridhar et al. [31] reported that incorporating JF at a rate of 1.5% by weight of cement in normal-strength concrete (NSC) resulted in a 18% net increase in CS and a 25% increase in FS. Another study [22] reported that the highest enhancements in CS were observed at a volume percentage of 0.25% of JF, whereas the maximum improvements in flexural–tensile strength were observed at a volume percentage of 0.5% of JF. There have been numerous studies that have confirmed the favourable impact of JF inclusion on the mechanical characteristics of concrete. However, the effects of JF on parameters related to durability have not yet been established. According to Gulzar et al. [6], the inclusion of JF had unfavourable effects on the properties of ordinary concrete related to permeability. As the volume of JF increases, the porous nature of plant fibres leads to a decline in the concrete's ability to resist water absorption and chloride ion penetration. The inclusion of SCMs as a part binder in concrete, such as fly ash or SF, can help in controlling the worse impacts of plant fibres on the durability of material [6,7].

Previous research showed that the absorption-related characteristics of JF-reinforced concrete have not been adequately addressed. JF is a low-cost and greener alternative to synthetic fibres, and it offers high tensile strength. Moreover, no reliable information was found on the flexural response and life cycle impact of JFRC. It is widely recognized that the alkaline environment of cement can cause corrosion of the components of plant fibres [32]. The addition of a secondary binding material, such as SF or rice husk ash, to cement has been shown to mitigate the corrosive impact of an 'alkaline' environment on the plant fibres. The "pozzolanic reactions" convert free calcium-hydroxide (CH) into calcium-silicate-hydrate (C-S-H) [21,33]. There is limited research available on the modification of JF-reinforced concrete using SF.

This study aimed to examine the impact of various JF volume percentages on the mechanical, durability, and life cycle aspects of concrete, with and without SF as an SCM. The study examined the mechanical properties of the mixtures, including CS, flexural

(load–deflection characteristics, ductility, residual strength), and tensile strength. This study conducted various tests to assess the durability behaviour of JFRCs, such as WA by immersion (%), RCIP/electric flux (Coulombs), ER, and UPV, to evaluate the durability of the modified JFRC mixtures. The findings of this research could provide useful knowledge on how to efficiently incorporate plant fibres in the design of concrete with improved flexural–tensile performance and resistance to weathering agents. JFRC possesses both economic and environmental significance. The use of JF as reinforcement reduces the cost of concrete production while potentially providing improved strength and durability. Additionally, JF is a renewable/reproduceable resource (unlike manufactured fibres), and their use in concrete helps to reduce carbon emissions, making it an eco-friendly alternative to traditional reinforcement materials. Besides, the natural synthesis of JF allows the absorption of $CO_2$ from the atmosphere. Therefore, plant-based/agro-fibres are suitable for a sustainable built environment.

## 2. Materials and Methods

### 2.1. Constituent Materials

As per ASTM C150 [34], Portland cement with a strength grade of 53 was utilized as the primary binder. The cement used in the study had a specific surface area (fineness) of 356 m$^2$/kg and an apparent density of 3.13 g/cm$^3$. Table 1 presents the oxide percentages of the cement used in the study. SF was also utilized as an SCM. Figure 1 displays an SEM image of the small particles of SF. The "apparent-density" and "surface-fineness" of SF was determined to be 2.33 g/cm$^3$ and 27,000 m$^2$/kg, respectively. Figure 2 shows the particle-size distribution of the binders used in the study.

**Table 1.** Oxide composition of OPC.

| Oxide | (%) |
|---|---|
| $SiO_2$ | 17.44 |
| CaO | 65.17 |
| $Al_2O_3$ | 4.61 |
| MgO | 2.16 |

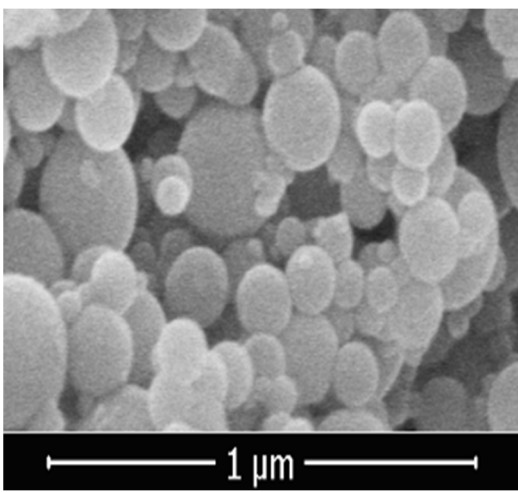

**Figure 1.** SEM results of SF sample.

This research used fine and coarse, for the mixtures. Siliceous sand with a "bulk-density" of 1.63 g/cm$^3$ was used as the "fine" aggregate, while crushed dolomite aggregate with a "particle-density" of '2.68 g/cm$^3$' was used as the "coarse" aggregate. Aggregates that were "fine" and "coarse" had a 24 h WA capacity of "0.98%" and "1.12%", respectively. The gradation test results are depicted in Figure 3.

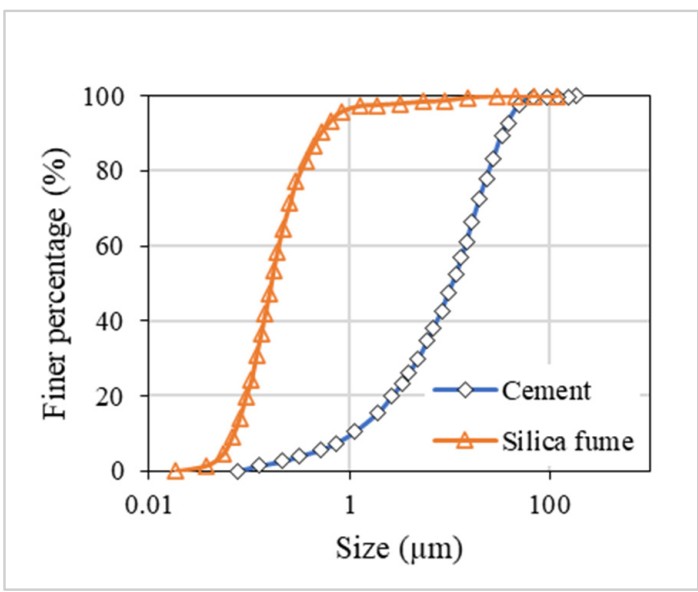

**Figure 2.** Particle size distribution of cement and SF.

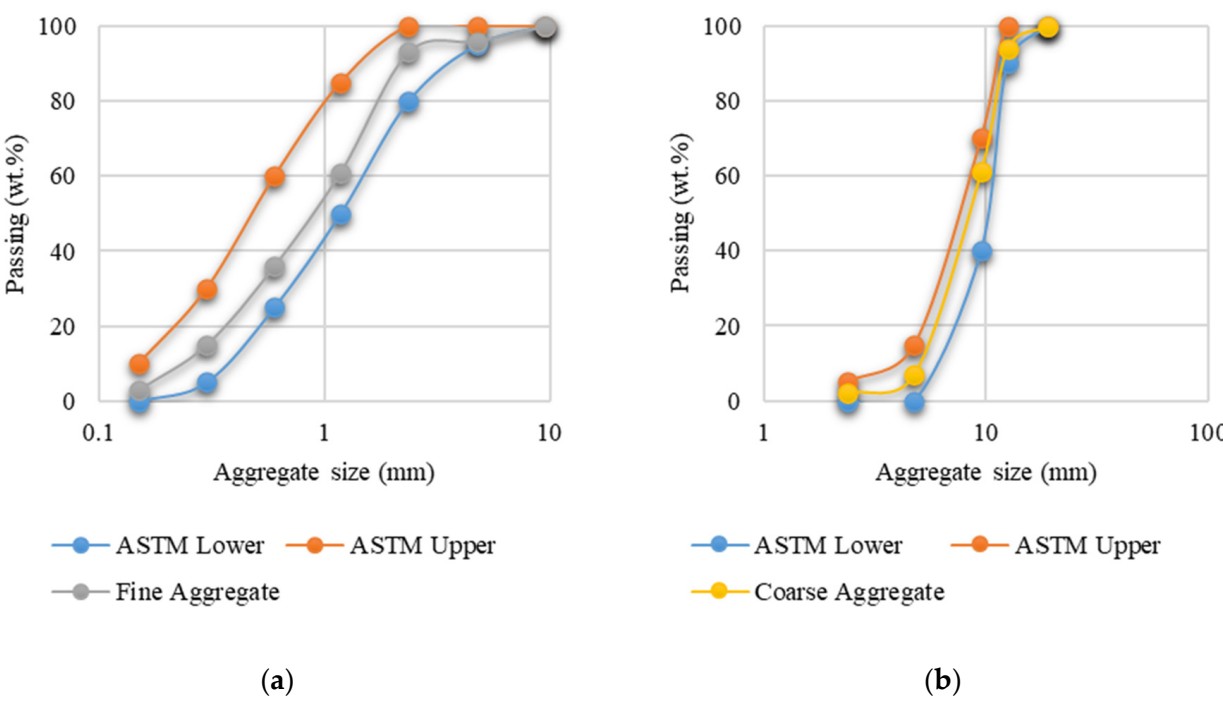

(**a**)                                                    (**b**)

**Figure 3.** The particle size distribution of aggregates. (**a**) Fine; (**b**) coarse.

The research involved the production of JF using jute sacks that were no longer usable. The sacks were converted into fibres with varying lengths between 12 and 18 mm. As a result, the fibres utilized in the study were classified as recycled fibres. The average length of 15 mm was chosen as the optimal length for JF based on previous research, which demonstrated similar lengths of JF resulted in improved mechanical performance [28]. It was discovered that the average diameter of JF was approximately 0.1 mm. JF had an apparent density of 1.45 g/cm$^3$, and its tensile strength was 410 MPa. The general characteristics of JF are depicted in Figure 4. To reduce the amount of water used and produce concrete with a low water-cement ratio (w/c), a polycarboxylate-based plasticizer was utilized.

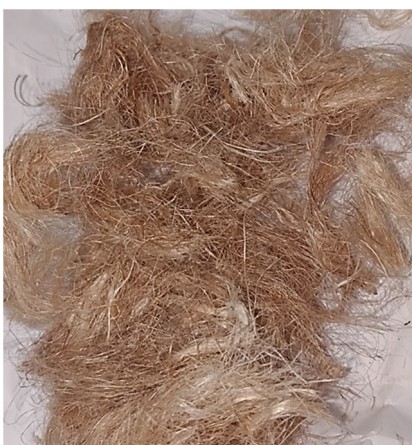

**Figure 4.** Jute fibre (JF).

### 2.2. *Design and Preparation of Concrete Mixtures*

Table 2 presents the names and proportions of the various components used in the mixture. Various trials were conducted to establish the optimal mixture ratio for the control concrete (JF 0) with the goal of attaining a CS of "60 MPa" and a slump range of $180 \pm 30$ mm. The mixtures had JF at three different volume percentages: 0.15%, 0.3%, and 0.5%. However, any volumes of JF higher than 0.5% were not considered as they have been reported to have negative effects on mechanical strength [22,28].

**Table 2.** Design of mixtures.

| Mix Names | JF (%) | Type I Cement (kg/m$^3$) | SF (kg/m$^3$) | Fine Aggregate (kg/m$^3$) | Coarse Aggregate (kg/m$^3$) | Water (kg/m$^3$) | SP (kg/m$^3$) | JF (kg/m$^3$) |
|---|---|---|---|---|---|---|---|---|
| JF0 | 0 | 550 | 0 | 640 | 1073 | 182 | 3.03 | 0.00 |
| JF0.15 | 0.15 | - | 0 | - | - | - | - | 2.16 |
| JF0.3 | 0.3 | - | 0 | - | - | - | - | 4.32 |
| JF0.5 | 0.5 | - | 0 | - | - | - | - | 7.20 |
| JF0/SF | 0 | 495 | - | - | - | - | - | 0.00 |
| JF0.15/SF | 0.15 | - | - | - | - | - | - | 2.16 |
| JF0.3/SF | 0.3 | - | - | - | - | - | - | 4.32 |
| JF0.5/SF | 0.5 | - | - | - | - | - | - | 7.20 |

Two sets of JFRCs were manufactured, with one using OPC only and the other replacing 10% of OPC with SF. The addition of SF to the mixture would serve two purposes: a "pozzolanic" effect and "mineral-filler" effect, as well as a reduction in the damaging impact of the highly alkaline cement on the plant fibre [35].

Figure 5 illustrates the process of mix preparation. The mechanical drum mixer was used to dry-mix the binders and aggregates for 4 min. During the next 4 min duration, water was added gradually to the dry mix along with the required amount of SP (at a rate of 3 litres per cubic meter). During the last stage of the mixing process, the fibre was slowly added to the wet mix over the course of 4 min. During all stages, the speed of mixing was around 20 rpm. After measuring the slump value of the fresh concrete, the mixture was then poured into moulds of different standard shapes. To ensure consistency across all mixtures, a vibrating table was used to compact the samples with a fixed vibration duration of 30 s. The compacted specimens were carefully shielded with a highly durable and waterproof membrane and then stored in a temperature-controlled environment for an exact duration of 24 h to ensure optimal setting conditions. After carefully removing the specimens from the moulds, they were placed in a water tank at a precisely controlled temperature of $25 \pm 3$ degrees Celsius to guarantee optimal curing conditions for the specified duration of testing.

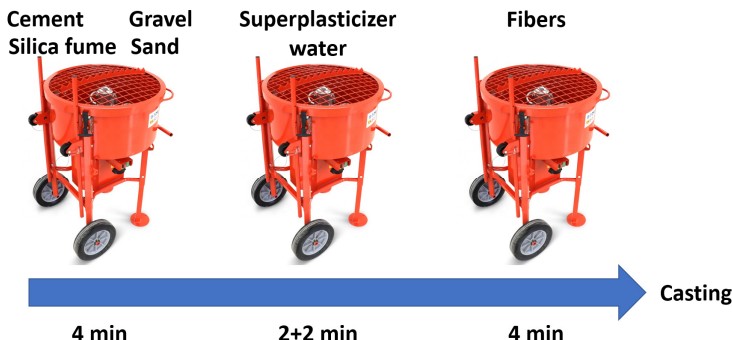

**Figure 5.** Mixing procedure.

### 2.3. Determination of Concrete Properties or Engineering Performance

Fresh concrete properties, namely fresh density, air content, and slump, were assessed. Air content and Abram's cone slump tests were conducted by following ASTM C231 [36] and ASTM C143 [37].

To determine the CS of the concrete specimens, "cubic" samples measuring 100 mm were tested according to the guidelines specified in the ASTM C39 standard [38]. The testing process involved applying a compressive force to the samples until they failed, and the maximum load that the specimens could bear before collapsing was recorded as the CS.

The STS of the concrete specimens was determined by testing "cylindrical" samples measuring "100 mm in diameter" and "200 mm in height" in accordance with the ASTM C496 standard [39]. The testing process involved applying a tensile force to the samples along the vertical axis until they failed. The maximum tensile load that the samples could withstand before failure was recorded as the STS or indirect tensile strength.

The FS of the concrete specimens was determined by testing "beam" samples measuring "100 mm in width", "100 mm in depth", and "350 mm in length", in accordance with the ASTM C1609 [40]. The testing process involved applying a load to the centre of the beam until it failed, and the maximum load that the beam could bear before collapsing was recorded as the FS. The load vs. deflection behaviour was also noted for specific mixtures (JF0.3, JF0.5, JF0.3/SF, and JF0.5/SF).

After 28 days of curing, the WA capacity of the concrete specimens was determined by immersing "disc-shaped" samples measuring "100 mm in diameter" and "50 mm in height" in water for a 24 h period in accordance with ASTM C948 [41]. The immersion method involved weighing the dry samples before and after immersion in water to determine the amount of water absorbed by the samples.

The 28-day RCIP values of the concrete specimens were determined by testing disc-shaped samples measuring "100 mm in diameter" and "50 mm in height", in accordance with ASTM C1202 [42]. This test is used to assess the ability of the concrete to resist the penetration of chloride ions, which can cause corrosion of reinforcing steel and affect the durability of the concrete structure. The RCIP test involves applying a voltage across the concrete sample and measuring the electrical charge/flux (Coulombs) passing through it.

The ER of the concrete mixtures was determined by testing cubic samples measuring "100 millimetres" on each side in accordance with ASTM C1876 [43]. The ER test is used to assess the ability of the concrete to resist the flow of electric current and is related to various properties such as permeability, durability, and corrosion resistance. In this test, a small electrical potential (60 V) is applied to the sample, and the resulting current is measured.

To evaluate the quality of the concrete, a UPV test was conducted. UPV values are known to be indicative of the concrete's strength, porosity, internal defects, and permeability. For this test, "100 mm cubic" samples were used, and the procedure followed ASTM C597 [44].

*2.4. Life Cycle Assessment or Environmental Performance*

The study evaluated the environmental impact (EI) of various concrete mixtures by analysing their global warming potential (GWP) resulting from carbon dioxide ($CO_2$) emissions. To assess the EI, the functional unit considered was one cubic meter ($m^3$) of concrete. Therefore, the analysis focused on determining the EI of producing one cubic meter of concrete.

The research investigated the EI of the materials used to make concrete, from when they are extracted from the Earth to when they are delivered to the concrete plant. Additionally, the study assessed the EI of the concrete production process, including the "extraction" of raw materials (B1), the "transportation" of processed materials (B2), and the actual "manufacturing" process (B3). In other words, the study aimed to analyse the environmental impact of the entire life cycle of concrete production, from start to finish. This kind of analysis is important for understanding the potential environmental impact of construction materials and processes and for identifying ways to reduce that impact.

To assess the EI resulting from the "processing" of "raw materials" (B1), the study utilized a database containing the GWP values (kg-$CO_2$/kg). These values were obtained from a previous study conducted by Braga et al. [45]. The EI of SF used in the concrete mixtures was obtained from a study by Hájek et al. [46]. The data for the EI of the raw materials used in the concrete mixtures are presented in Table 3. The collected data were adapted based on the cradle-to-gate scenario. The EI of JF was taken as similar to that of the recycled coconut fibre, since, in this research, JF was considered as a recycled material [47]. The study considered the environmental impact of transporting the concrete mixtures by lorry, which was calculated to be $6.57 \times 10^{-5}$ kg-$CO_2$ per kilogram per kilometre. Table 4 provides information on the distances between the "concrete plant" and the "sources of raw materials", which were used to determine the EI of transporting the raw materials. The distances were utilized to evaluate the EI of the constituent materials' transportation.

**Table 3.** EI of raw materials (kg-$CO_2$/kg).

| Material | GWP (kg-$CO_2$/kg) |
|---|---|
| OPC | 0.898 |
| Siliceous sand | 0.002 |
| Dolomite sandstone | 0.053 |
| SP | 0.002 |
| Water | 0.000 |
| SF | 0.0011 |
| JF | 0.36 |
| Concrete preparation | 4.65 |
| Transportation impact | 0.0000657 |

**Table 4.** Transportation distances.

| Material | Distance from Mixing Plant (km) |
|---|---|
| OPC | 397 |
| SF | 178 |
| Siliceous sand | 456 |
| Dolomitic sandstone | 246 |
| SP | 177 |
| JF | 15 |
| Water | 0 |

## 3. Results and Discussion

*3.1. Engineering Performance*

3.1.1. Air Content

The effect of JF and SF incorporation on the air content of fresh concrete is illustrated in Figure 6. It can be noticed that, for both SF- and non-SF-containing mixtures, the "air

content" increased with the increasing percentage of JF. The addition of plant-based fibres to concrete can increase its air content due to the interaction and cohesion between the fibres and the fresh concrete mixture. The fibres act as obstacles, hindering the movement or flow of the fresh concrete mixture, which in turn can create voids in the mix. These voids can trap air and increase the overall air content of the concrete. The lack of workability and balling effect caused by the presence of JF increases the air content in the compacted concrete. Similar findings have been reported by the use of coconut fibre in fresh concrete [48]. The inclusion of SF as the cement replacement material marginally decreases the air content of fresh concrete. This can be attributed to the filling effect of extremely fine SF particles that occupy the free spaces between cement particles, thus resulting in a more compacted concrete.

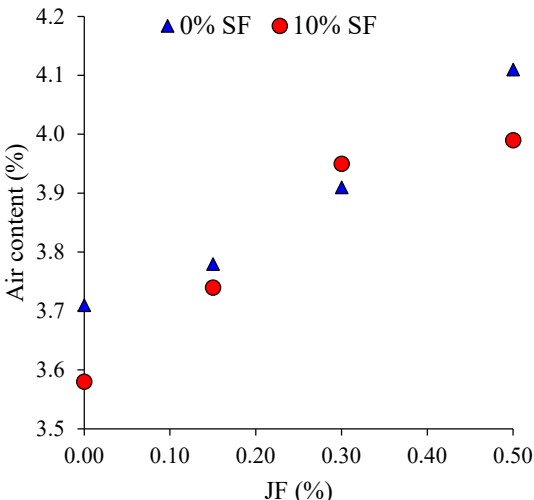

**Figure 6.** Effect of JF and SF incorporation on the air content of fresh concrete.

### 3.1.2. Fresh Density

The results of the fresh density tests are presented in Figure 7. The fresh density of concrete is of great practical significance since it is widely used to assess the quality and yield. It can be noted that plant-based JF reduces the fresh density of concrete with the increasing percentage. Up to a 1% decline in the fresh density of concrete was observed when the JF content was used up to a 0.5% concentration. This is primarily linked to the low density of JF filaments as compared to the binder matrix. Since low-density JF filaments replace the dense cementitious matrix in the concrete, it is not possible to control the loss in density due to the incorporation of the lightweight fibres.

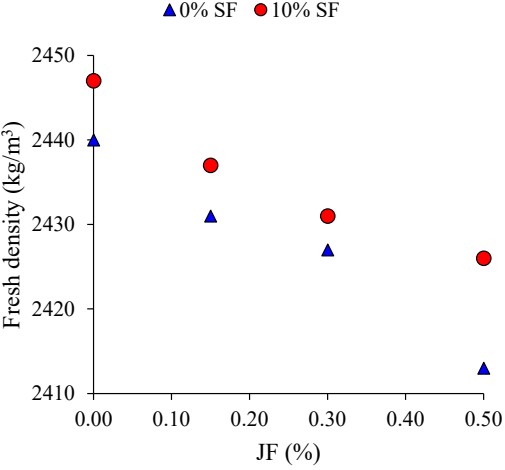

**Figure 7.** Effect of JF and SF incorporation on the density of fresh concrete.

Contrary to JF incorporation, SF addition increased the fresh density of concrete by 7 kg/m$^3$. As SF particles are extremely fine as compared to OPC particles, therefore, these can occupy the volume between OPC particles, resulting in increased compaction and densification. SF is a highly reactive pozzolan that can improve the packing density of the concrete particles, leading to a denser mixture [49]. The fine particles of silica fume can fill the voids between the larger particles of the other components, resulting in a more compact mixture.

### 3.1.3. Slump

The effect of JF and SF addition on the workability of fresh concrete is depicted in Figure 8. The addition of JF to a concrete mixture can generally make it more viscous and less fluid, reducing its workability due to possible clustering of thin JF filaments. JF incorporation can increase the friction between concrete particles, making it harder to mix and move the concrete. However, it can be noted that, despite the decline in the workability, the slump value of all fibre-reinforced mixes remained within the target slump range of 150–210 mm. This is because the control fresh concrete was produced with a carefully designed dosage of a high-range water reducer, which allowed for the possible workability loss due to the incorporation of JF. The concrete yielding a slump in the range of 150–230 mm can be used as a pumpable concrete.

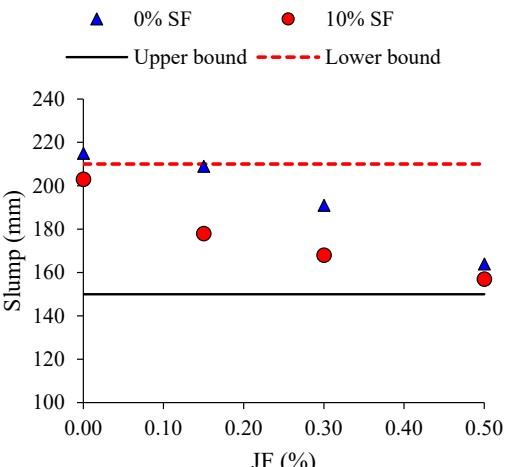

**Figure 8.** Effect of JF and SF incorporation on the slump of fresh concrete.

The incorporation of SF led to a small decline in the workability of the fresh concrete. The addition of SF to a concrete mixture can have a negative impact on workability due to its high surface area and fineness as compared to OPC particles, which can cause the concrete to become thicker, cohesive, and less fluid. SF particles can absorb water, resulting in a reduction in the water content available for cement hydration and increasing the demand for water. Therefore, the use of SP provides control over the significant loss of workability due to SF addition. SP does not only control the negative effects of SF and JF incorporation on the workability, it also helps in the dispersion of SF particles and JF filaments to ensure more homogenous and enhanced properties of the concrete [50].

### 3.1.4. Compressive Strength

Figure 9 depicts how the CS of concrete is influenced by inputs such as the presence of varying percentages of JF and SF. The impact of different volumes of JF on the CS of concrete was not consistent. At 28 days, there was a slight increase of "4.0%" and "4.8%" in CS when "0.15%" and "0.3%" volumes of JF were added, respectively. On the other hand, the CS of concrete at 91 days showed a higher increase. There was an increment of "5.5%" and "9.7%" when "0.15%" and "0.3%" of JF were added, respectively. The reason for the increase in CS when 0.15% to 0.3% of JF was added is due to the enhancement of the

"transverse" distortion resistance of the plain concrete [22,51]. There is also a theory that the moist JFs can aid in the internal curing of the concrete's microstructure, which may lead to an increase in its strength [3]. Due to the lower density and higher surface area of JFs, their incorporation at higher rates may have a detrimental effect on the density of concrete. Therefore, when a 0.5% volume of JF was added, there was no significant change observed in the CS at both 28 and 91 days of testing.

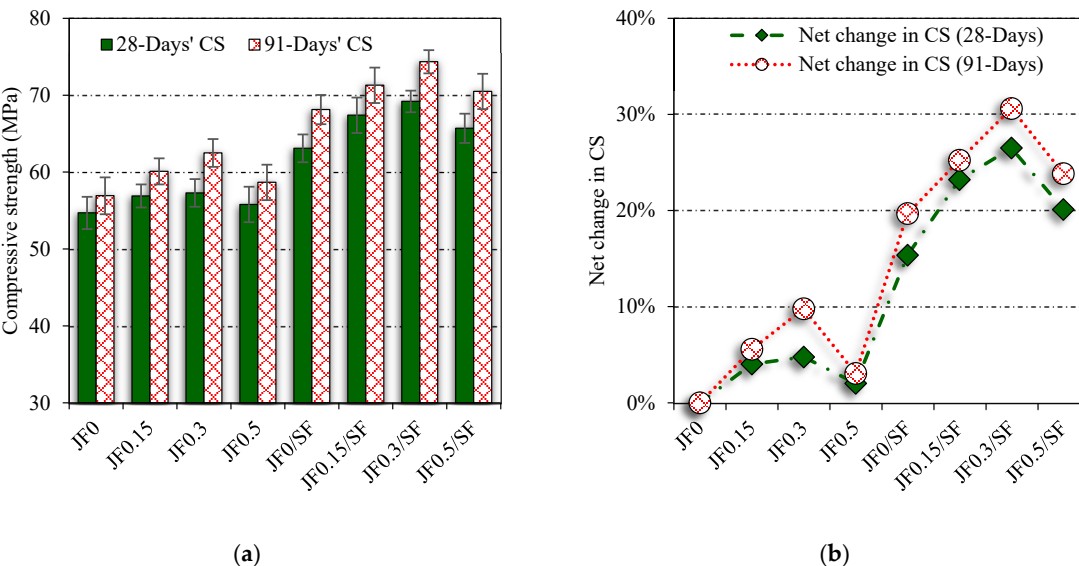

**Figure 9.** Compressive strength results: (**a**) CS results with SF and varying JF content; (**b**) change in CS with respect to JF0.

When a 10% volume of SF was added to the concrete, there was a significant increase in the CS. At 28 and 91 days, there was an increase of 15.4% and 19.7% in CS, respectively. The reason behind this increase in strength was attributed to the pozzolanic reaction between the portlandite (CH) and SF particles, which led to the additional growth of C-S-H in the concrete. SF contributes to the increase in strength, not only through the pozzolanic reaction, but also by reducing the pores between cement particles, which leads to the solidification and strengthening of the microstructure.

When JF and SF were incorporated together, their combined effect showed a synergistic increase in the CS of the concrete. When a 0.3% volume of JF and a 10% volume of SF were added individually, there was a net increase of 5% and 15%, respectively, in CS at 28 days. However, when they were added together, there was a synergistic effect, which resulted in a much greater net increase of 26% in CS. When a 10% volume of SF and a 0.3% volume of JF were added together, there was a significant increase in CS compared to the control mixture. At 28 and 91 days, there was a net increase of 26% and 31%, respectively. The observation of similar synergistic effects between SF and other volumes of JF suggests that the efficiency of JF as a fibre reinforcement increases as the strength grade of PC improves. The enhancement in the fibre–matrix interfacial strength due to the incorporation of SF in the binder matrix may explain this phenomenon [52]. An important point to mention is that the increase in CS resulting from the addition of JF also increased as the concrete aged. For example, with the addition of 0.3% JF, the strength of the concrete increased by 4.8% and 9.8% (compared to the control mixture) at 28 and 91 days, respectively. This indicated that the strength improvement due to JF incorporation continues to increase with the aging of the concrete (not indefinitely).

### 3.1.5. Splitting Tensile Strength

The impact of adding JF and SF on the STS of concrete mixtures is presented in Figure 10. The figure shows that the inclusion of JF resulted in significant increases in

the STS value of concrete. At different volumes of incorporation of JF, there were notable increases in the STS value of concrete, as seen in Figure 10. Specifically, there were net increases of "8.6%", "18.4%", and "19.8%" in the STS value of the concrete at "0.15%", "0.3%", and "0.5%" JF incorporation, respectively. The strength values of the concrete also showed notable improvements with an increase in JF content, particularly in the 91-day STS. Islam and Ahmed [22] reported that the addition of 0.5% of JF led to a net increase of approximately 20% in the splitting tensile strength of normal-strength concrete. Zakaria and colleagues [28] reported that the addition of 0.15–0.25% of JF resulted in a 16–20% improvement in the STS of PC. The incorporation of JF in concrete is believed to enhance the resistance of concrete against the onset and propagation of cracks. This can be explained by the fact that JFs can bridge the micro-cracks in the concrete matrix, which can lead to an increase in the crack-bridging capacity of the concrete. As a result, the stress concentration around the cracks is reduced, which can prevent the further growth and development of cracks. This mechanism ultimately leads to an improvement in the tensile strength of the concrete. Additionally, the improved interfacial bonding between the fibres and the cement matrix due to the pozzolanic reaction can also contribute to the increased tensile strength of the concrete [53]. The study found that the highest STS value was observed in the concrete containing a 0.5% volume of JF. However, it was noted that the percent difference in the STS value between the concrete mixtures containing a 0.3% and a 0.5% volume of JF was not significant. In other words, while increasing the volume percentage of JF resulted in higher STS values, the improvement in STS between 0.3% and 0.5% was not as significant as the improvement seen between lower percentages of JF. This suggests that the optimal volume percentage of JF may depend on various factors, such as the specific application and required strength of the concrete. Based on the results, it can be concluded that incorporating 0.3% vol. of JF in the mixture can provide optimum benefits in terms of the efficient utilization of fibre material and maintaining the workability of the fresh mixture. Therefore, this volume percentage can be considered as the ideal amount of JF to be added to the concrete mixture.

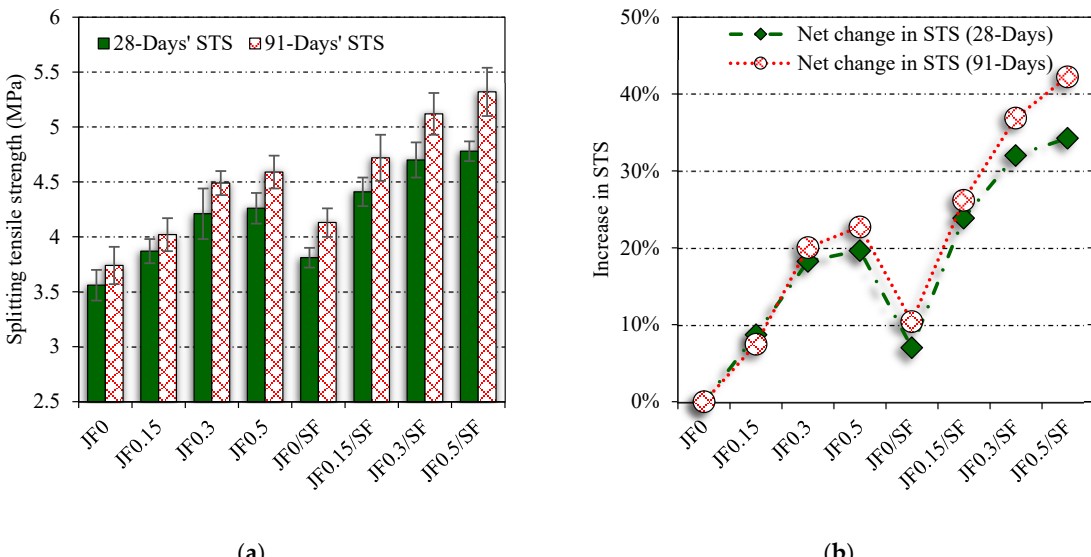

(a)　　　　　　　　　　　　　　　　　　　　　　(b)

**Figure 10.** Splitting tensile test results: (**a**) STS results with SF and varying JF content; (**b**) change in STS with respect to JF0.

At 28 and 91 days, the strength of concrete was improved by 7% and 10.4%, respectively, compared to the control mixture, when SF was added to it. When both JF and SF were added to the concrete mixture, their combined effect showed a coupling and synergetic effect on the STS of the concrete. This means that the combined effect was greater than the sum of the individual effects of JF and SF. In other words, the use of JF and SF together resulted in a greater improvement in the STS of the concrete than using either one of them

alone. This highlights the potential benefits of using multiple materials to enhance the properties of concrete. The combination of JF and SF in the concrete mixture resulted in a coupling and synergetic effect on the STS. Specifically, when 0.3% JF and 10% SF were added together, there was a net improvement of 32% in STS (which was 7% higher than the improvement achieved by using SF alone) at 28 days, and a net improvement of 42.2% in STS (which was 9% higher than the improvement achieved by using SF alone) at 91 days. This showed that the combined use of JF and SF can lead to significant improvements in the short-term and long-term strength of concrete. Adding SF to the concrete mixture enhances the pull-out strength of fibre filaments [52]. The higher STS of the concrete was a result of the formation of C-S-H and an improved JF–matrix interface, which led to an increase in the micro-hardness and a stronger bond between the fibres and the matrix.

### 3.1.6. Load versus Deflection

The effect of JF and SF incorporation on the load–deflection behaviour of the concrete is shown in Figure 11. It is distinct that the incorporation of SF had a slight influence on the peak load and no contribution towards the post-peak deflection behaviour of the concrete. Adding SF contributed to the strength improvement through pozzolanic reactions by creating more C-S-H gels. However, the incorporation of JF significantly changed the load–deflection behaviour of the concrete. JF0.3 and JF0.5 generally demonstrated greater ductility compared to traditional PC, allowing them to withstand more extensive deformations before collapsing. This is because the JF filaments distributed within the concrete help to distribute loads more evenly and bridge small cracks that could form under external loads [54].

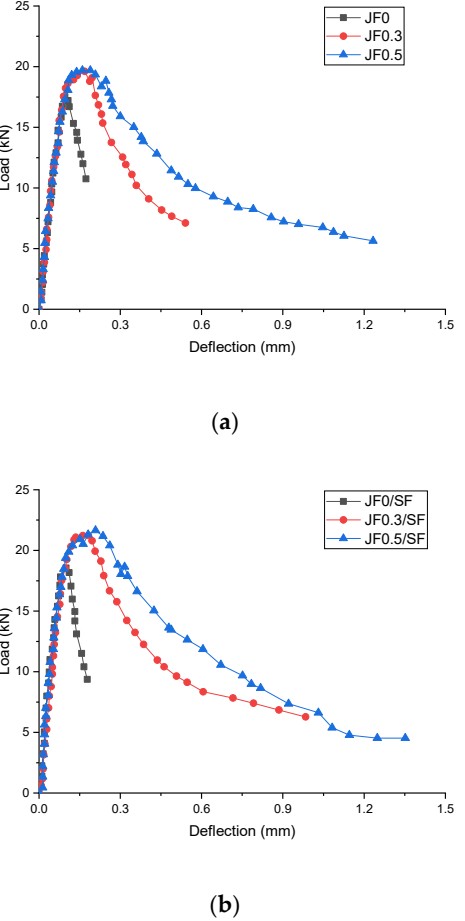

(**a**)

(**b**)

**Figure 11.** Load deflection behaviour of concrete with (**a**) different JF contents and 0% SF and (**b**) different JF contents and 10% SF.

The load–deflection curve of JFRC may exhibit plateaus or yielding, which correspond to the formation of new cracks and fibre pull-out events. The final failure of JFRC may occur due to the rupture of JF filaments or the complete separation of JF filaments from the matrix. The increase in JF content had a minor improving effect on the peak load sustained by JFRC. However, the increase in the fibre content ensured the delay in the failure of JFRC. This is because, at higher fibre contents, more fibres are present to restrict the crack movements and allow for a more ductile response than concretes with low fibre contents.

The incorporation of SF into JFRCs led to a further increase in the peak load and improved the post-peak deflection behaviour. The longer stems of the post-peak load–deflection were observed for JFRC mixtures incorporating SF. This can be attributed to the improvement in the bond performance of JF due to the solidification and densification of the concrete matrix with SF addition. Thus, SF can help improve the overall load–deflection behaviour of JFRCs.

### 3.1.7. Flexural Strength or Modulus of Rupture

Figure 12 presents a visual representation of how incorporating JF into concrete, with and without SF, affects the FS of the material. Adding a 0.3% and a 0.5% volume of JF to the concrete resulted in a net increase of approximately 16% in the FS of the material after 28 days. The addition of a 0.3–0.5% volume of JF to the concrete resulted in a net improvement of approximately 20% in the flexural strength (FS) of the material after 91 days. Concrete samples showed further improvements in strength at later ages. As the concrete aged, the chemical reactions within the mixture continued to occur, resulting in the formation of stronger bonds between the cement particles and other materials including the fibre reinforcement within the mixture. This finding corroborates the findings of a prior investigation [55], which revealed that the increase in FS resulting from the addition of JF can be attributed primarily to the reinforcement of the concrete against the initiation and propagation of cracks. The incorporation of JFs helps to distribute stress more uniformly throughout the concrete matrix, which reduces the likelihood of the formation of cracks and other forms of damage. By strengthening the concrete and reducing the risk of cracking, the JFs helped to increase the FS.

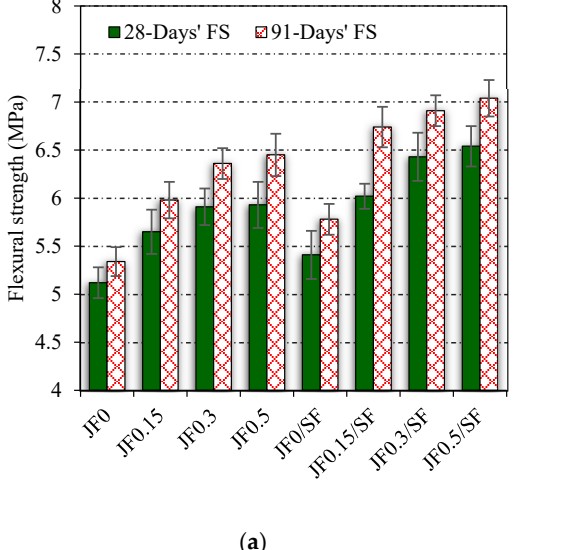
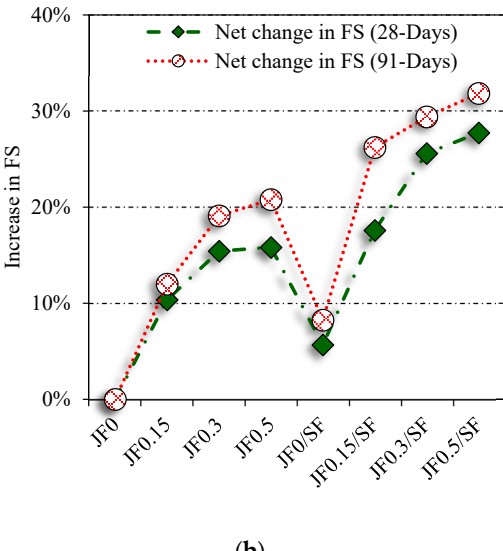

(**a**)          (**b**)

**Figure 12.** Flexural test results: (**a**) FS results with SF and varying JF content; (**b**) change in FS with respect to JF0.

It is worth noting that the difference in the percentage of improvement in FS resulting from the addition of a 0.3% and a 0.5% volume of JF was not significant. Hence, a 0.3% volume of JF may be regarded as the optimum amount to be added to the concrete mixture for maximum FS enhancement.

The FS of the concrete exhibited a net improvement of 5–10% as a result of incorporating SF as a partial substitute for OPC. In addition to enhancing the performance of OPC, the inclusion of SF also improved the impact of JF on the FS of the concrete. The joint addition of SF and a 0.3% volume of JF resulted in net improvements of 26% and 30% in FS at 28 days and 91 days, respectively, compared to the control mixture. The combined inclusion of JF and SF in the concrete mixture exhibited both a coupling effect and a synergetic effect on the FS of the material. The addition of SF had the effect of refining the bond between the fibre and binder matrix, which increased the pull-out strength of the fibre filaments. This, in turn, led to additional enhancements in the tensile and flexural strength of the concrete. Previous research has demonstrated the synergistic effects of incorporating SF into the concrete mixture with both micro and macro fibres. These findings suggest [56,57] that the inclusion of SF in combination with other reinforcing materials, such as JF, can lead to significant improvements in the strength of concrete, making it more resistant to cracking and other forms of damage.

### 3.1.8. Residual Strength and Flexural Toughness

The residual flexural strength (RS) and flexural toughness (FT) of concrete mixtures are presented in Figures 13 and 14, respectively. RS refers to the flexural strength of FRC after the initial cracking of the concrete. It is usually measured from the loads corresponding to 0.5 mm, 1 mm, and 2 mm deflections. However, in this study, JFRCs failed before 2 mm deflection; therefore, the RS measurement was limited to 0.5 mm and 1 mm only. The addition of JF filaments helps to bridge micro-cracks, preventing the cracks from propagating further and leading to a more gradual loss of strength after cracking. As a result, JFRC exhibited a higher RS compared to JF0. Due to the addition of 0.3% and 0.5% JF, the RS (0.5 mm) were 2.30 and 3.28 MPa as compared to 0 MPa RS of the plain JF0, respectively. Despite a nominal effect on the peak FS, the incorporation of 0.5% JF had a prominent effect on the RS at 1 mm deflection. JF0.3 and JF0.3/SF failed before 1 mm deflection, while JF0.5 and JF0.5/SF retained a notable RS at 1 mm deflection.

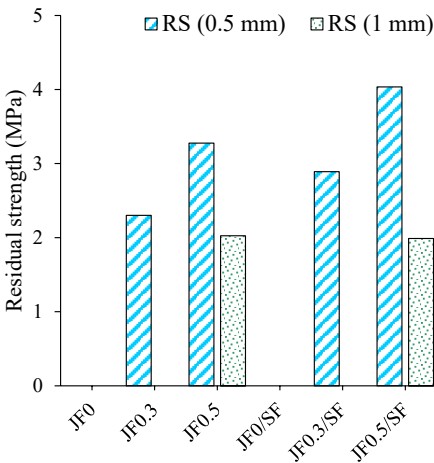

**Figure 13.** Effect of JF and SF addition on the residual strength.

FT refers to the ability of a concrete to absorb energy under "flexure" or "bending" and is an important property for many structural applications. JFRCs exhibit increased FT due to the bridging effect of the fibres, which redistributes stress around the crack and prevents further crack propagation. This results in a higher energy absorption capacity and increased ductility, allowing the JFRC to deform more before failing. For instance, JF0.3 and JF0.5 yielded 3.22- and 6.4-times higher FT as compared to JF0, respectively. The incorporation of SF also increased the FT of JFRCs. For instance, JF0.3/SF and JF0.5/SF showed FT values 5.35- and 7.31-times higher as compared to JF0, respectively.

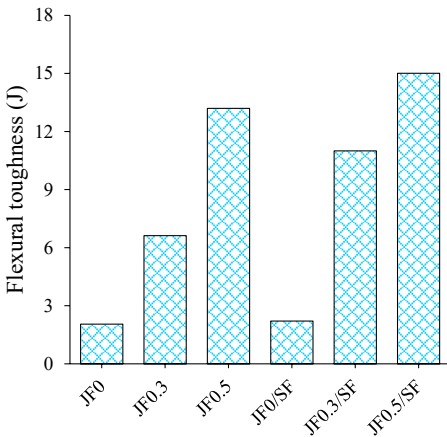

**Figure 14.** Effect of JF and SF addition on the flexural toughness.

3.1.9. Water Absorption Capacity

The durability of concrete is impacted by the movement of fluids through the pores that are connected to the concrete's surface. The volume of permeable voids, known as WA, can be used as an estimate of the durability of concrete. Essentially, the more permeable voids there are, the less durable the concrete is likely to be. Therefore, controlling the number of permeable voids in the concrete can help improve its durability. A test was carried out to measure WA, which represents the volume of permeable voids in concrete, at two different time intervals: 28 and 91 days. The results of the test are presented in a graph and figure labelled as Figure 15. This figure provides a visual representation of how WA changes over time due to the mixture design inputs such as JF volume and SF incorporation.

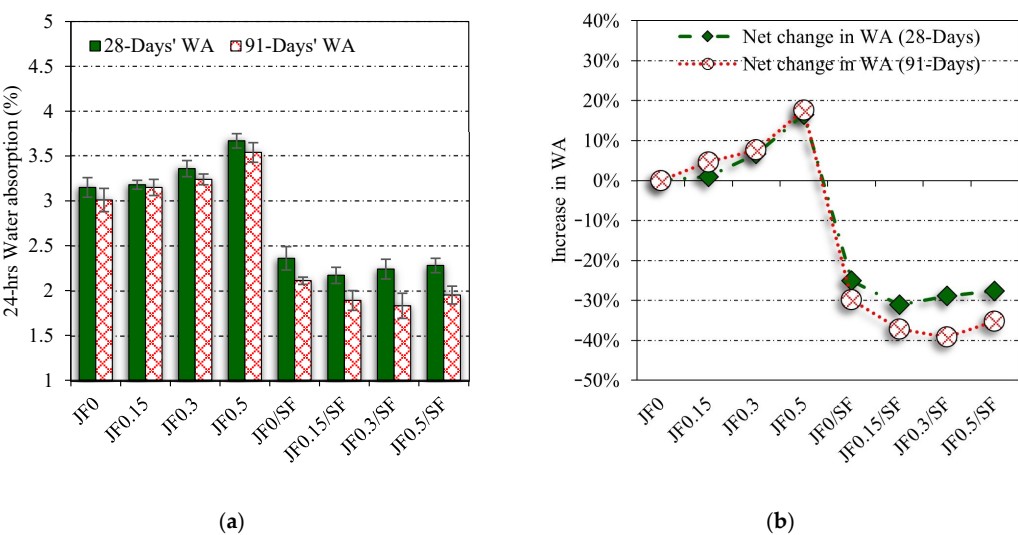

(a)                                                      (b)

**Figure 15.** Water absorption test results: (**a**) WA results with SF and varying JF content; (**b**) change in WA with respect to JF0.

The observation was made that, when the volume of JF was increased from 0.15% to 0.5%, the volume of permeable voids in the concrete (represented by WA) increased by approximately 16%. This suggests that the addition of JF may have a negative impact on the durability of the concrete by increasing its permeability. The increase in WA observed when JF was added to concrete is believed to be caused by an increase in the connectivity of the microstructure of the concrete with its outer surface. In other words, the addition of JF may lead to a more porous concrete microstructure, allowing fluids to move more easily through the material and increasing its permeability [12]. As the number of fibre filaments

in the concrete increased, more access points were created on the surface of the material, which can facilitate the movement of water into the concrete matrix.

The addition of SF resulted in a significant decrease in the volume of permeable voids, as represented by the WA capacity, in both plain and fibre-reinforced concrete mixtures. Specifically, the WA capacity decreased by 25.1% and 29.9% at the ages of 28 and 91 days, respectively. These results demonstrated that the incorporation of SF can be an effective way to improve the durability of JFRCs by increasing their resistance to the transport of fluids through their microstructure. The micro-particles of SF are believed to arrange themselves between the cement particles in the concrete mixture, which helps to densify the microstructure of the material. This can lead to a reduction in the volume of permeable voids and a corresponding decrease in the concrete's permeability. Additionally, the increased growth of calcium-silicate-hydrate ( C-S-H ) that occurred when SF was added to the mixture can also contribute to a reduction in the connectivity between capillary channels, further reducing the permeability of the concrete. Specifically, the WA values of the fibre-reinforced concrete containing JF (JFRC) mixtures were 30–40% lower than those of the control mixture.

### 3.1.10. Rapid Chloride Ion Permeability

Figure 16 displays the results of rapid chloride ion permeability (RCIP) testing conducted on various concrete mixtures. It can be observed that the RCIP values of the control concrete fall within the range of 1000 to 2000 Coulombs. RCIP testing is often used to assess the durability of concrete by measuring its resistance to the penetration of chloride ions, which can contribute to corrosion and other forms of damage in concrete structures. The moderate range of chloride permeability exhibited by the control concrete, as indicated by the RCIP values falling within the range of 1000 to 2000 Coulombs, is typically associated with concrete that has a low water-binder ratio, typically less than 0.4 [58]. These RCIP values are typically associated with high-strength and high-performance concretes [59]. The results indicated that there was a notable increase in RCIP with the increase in JF content. Specifically, when a 0.5% volume of JF was incorporated into the mixture, there was an observed increase in the RCIP value of approximately 43% compared to the control mixture. The observed increase in chloride permeability with the addition of JF can be attributed to an increase in the permeable porosity of the concrete, which allows for faster movement of chloride ions. As previously discussed, plant-based fibres such as JF are inherently porous in nature, and their incorporation into the concrete matrix can negatively impact its imperviousness [7,48].

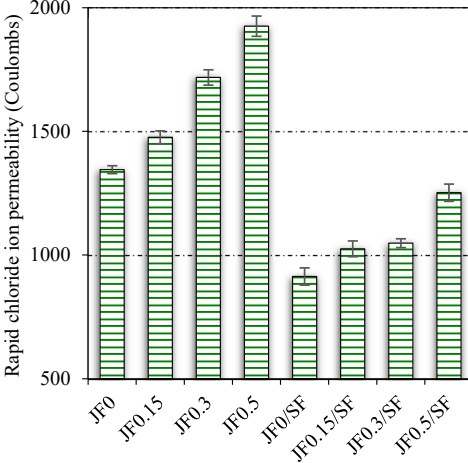

**Figure 16.** RCIP of studied concrete mixtures.

The addition of SF was found to be effective in reducing the RCIP value of the concrete. The observed reduction in the RCIP value with the incorporation of SF can be attributed

to its superior filler effect, which helps to control the free movement of chloride ions in the concrete matrix. This is because SF has been shown to enhance the chloride binding capacity of the microstructure, effectively reducing the permeability of the concrete to chloride ions. The results indicated that the addition of SF led to a significant decline in the RCIP value of the concrete, with a reduction of approximately 32% compared to the control mixture without SF. Previous research [59,60] has shown that SCMs such as SF and rice husk ash have a significant effect on the permeability-related properties of concrete. The study conducted by Kou et al. [59] reported a reduction of approximately 25% in the RCIP value of concrete with the incorporation of 10% SF. The negative impact of JF on the chloride permeability resistance of concrete can be mitigated by incorporating SF into the mixture. By adding SF to the JFRC mixtures, lower RCIP values were obtained compared to the control mixture. This implies that the use of SF can be an effective strategy for improving the durability and resistance of fibre-reinforced concrete to chloride ion penetration, which can enhance its lifespan and performance in various applications.

3.1.11. Ultrasonic Pulse Velocity

UPV, or ultrasonic pulse velocity, is a non-destructive technique used to evaluate the quality and integrity of concrete. It is used to detect any cracks or defects present in the concrete and to determine the homogeneity of the hardened concrete. UPV is a valuable tool for assessing the overall condition of concrete structures without causing any damage. The range of UPV values between 3.5 km/s and 4.5 km/s is generally associated with normal-strength or medium-strength concretes of good quality [61]. These values indicated a relatively homogeneous and sound concrete structure without any significant cracks or defects. To clarify, UPV values above 4.5 km/s are typically associated with high-strength concrete and ultra-high-performance concretes (UHPCs) that have a low water-binder ratio, indicating excellent quality. The UPV test results are presented in Figure 17, and as expected, the incorporation of JF had a negative impact on the UPV value. For example, when the JF content was increased from 0.15 to 0.5%, the UPV value decreased by 2.31%. The decrease in UPV due to the addition of plant-based fibres is a common phenomenon as these fibres have a lower density and cellular microstructure, which lead to a reduction in the overall density of concrete [12,62]. The UPV test results confirmed that the high fibre contents had a negative impact on the WA, RCIP, and CS results of the concrete. This is because the presence of plant-based fibres in concrete reduces its density and homogeneity, leading to a decline in the UPV values.

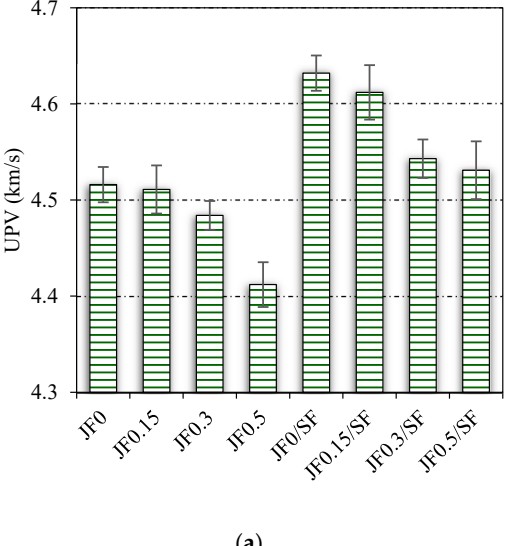

(a)

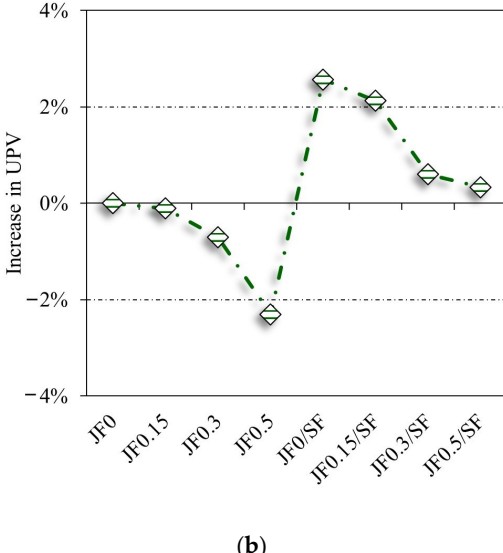

(b)

**Figure 17.** UPV test results: (**a**) UPV results with SF and varying JF content; (**b**) change in UPV with respect to JF0.

Adding SF increased the UPV value of the concrete by 2.6%. This increase can be attributed to the filler effect of SF, which helps improve the connectivity and homogeneity of the concrete. This, in turn, leads to a higher UPV value, which is above 4.5 km/s and is typically associated with excellent-grade concrete systems that have a low water-binder ratio, such as high-strength concrete. The compactness and reduction in pore size of the cementitious matrix around the fibres offsets the reduction in overall density caused by the presence of fibres in concrete [63].

### 3.1.12. Electrical Resistivity

ER is a crucial factor in determining the durability of concrete since the risk of corrosion in reinforced concrete is largely dependent on it. Figure 18 shows the ER measurements of all the concrete mixtures. Adenaert [64] suggested that, if the ER value of concrete is lower than 50 $\Omega$m, then the steel reinforcement is at high risk of corrosion. However, if the ER value ranges between 50 and 120 $\Omega$m, then the concrete is likely to be affected by corrosion in the long run. ER values above 120 $\Omega$m are indicative of concrete with very low corrosion risk. The results indicate that all concrete mixtures had ER values greater than 50 $\Omega$m. However, the PC mixtures exhibited higher ER values than their respective fibre-reinforced mixtures. The results showed that both plain concrete mixtures, with and without SF, are considered as durable and capable of protecting the steel reinforcement against corrosion. It should be noted that the incorporation of 10% SF into PC led to a 60% increase in ER. The increase in ER values of concrete containing SF can be explained by the filling of pores with micro-silica particles and hydration products. The C-S-H gel, which contributes significantly to the strength of concrete, reduces the formation of capillary pores, increases the volume of solid phases inside the microstructure, and ultimately, enhances the durability of concrete, including its resistance to corrosion [65]. The results indicated that the incorporation of JF has a negative impact on the corrosion-resistance potential of concrete, as evidenced by the results of WA and RCIP testing. Fibre-reinforced mixtures are more susceptible to corrosion due to the ability of the pore-solution to flow through the porous cellular internal structure of the fibres. Concrete mixtures with a 0.15% volume of JF, both with and without SF, have ER values above 120 $\Omega$m, which are considered safe against corrosion. The results obtained from the durability tests, namely WA, RCIP, and ER, highlight the importance of SF in preserving the overall durability of JFRC.

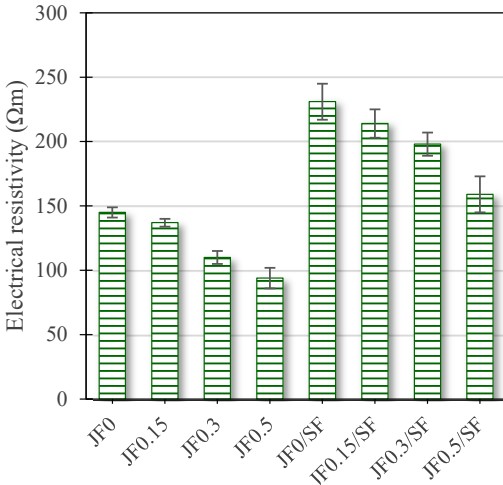

**Figure 18.** ER results of concrete mixtures.

### 3.2. Life Cycle Assessment Results

The global warming potential (GWP) of all studied concrete mixtures is illustrated in Figure 19. It can be observed that the incorporation of JF had a slight impact on the cradle-to-gate GWP of the concrete. Negligible reductions were noticed in GWP due to the increasing incorporation of JF. This is because JF possesses a very low GWP itself, and it replaces some volume of aggregates in concrete, which results in the marginal decline of

GWP. Contrary to JF inclusion, the incorporation of 10% SF as a cement substitution resulted in the 8% decline of the total GWP. Replacing OPC with SF in concrete can have a positive EI in several ways. SF is a by-product of the production of silicon and ferrosilicon alloys, and using it in concrete can help to reduce waste and limit the need for disposal. Additionally, the production of SF requires significantly less energy compared to cement production, resulting in reduced carbon emissions and energy consumption. By replacing a portion of the OPC with SF in concrete, the overall cement content of the mixture can be reduced. This can reduce the GWP of concrete as cement production is a significant contributor to greenhouse gas emissions (GHEs). Furthermore, SF can improve the durability and strength of concrete, allowing for thinner and lighter structures, which can further reduce the EI associated with construction.

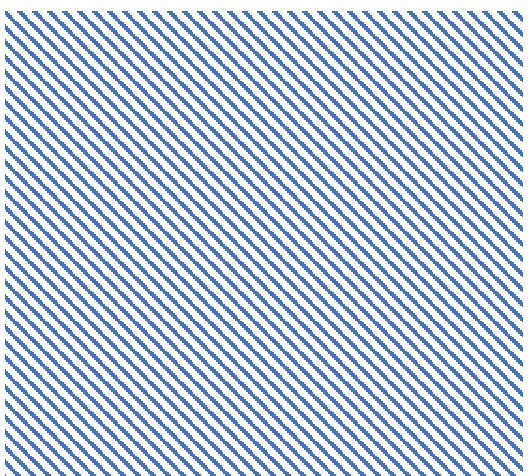

**Figure 19.** GWP of concrete mixtures with JF and SF incorporation.

The EI per unit CS, STS, and FS of each mixture is presented in Figure 20. These results show how much GWP is caused by the unit strength gained by each mixture studied. According to the results, it can be observed that the GWP per unit strength reduced with the rising JF content. GWP/STS and GWP/FS were highly sensitive to the JF addition, whereas GWP/CS was mildly influenced by the presence of JF. This is because JF yielded more benefit in the case of STS and FS, and it was mildly effective at enhancing the CS. On the other hand, SF addition was highly effective at reducing the GWP of concrete for unit CS, FS, and STS. The addition of 10% SF reduced the STS- and FS-related GWP by 16% and 14%, respectively, whereas JF0.5/SF yielded STS- and FS-related GWP corresponding to being 34% and 30% lower as compared to JF0.

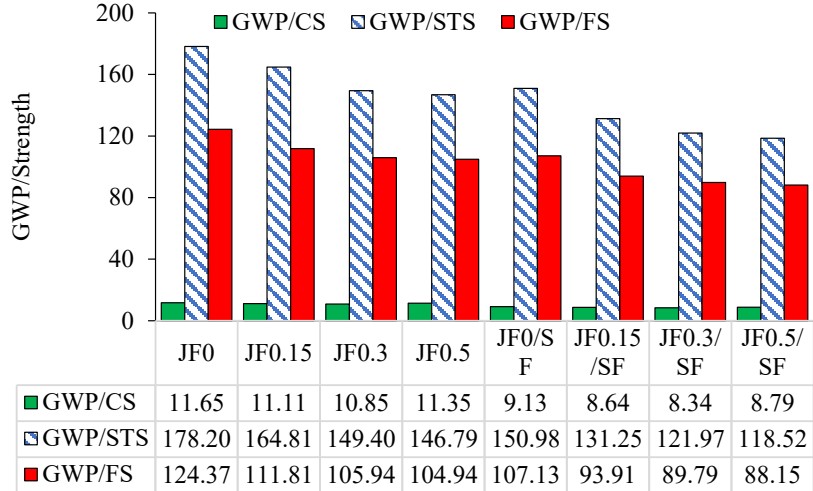

| | JF0 | JF0.15 | JF0.3 | JF0.5 | JF0/SF | JF0.15/SF | JF0.3/SF | JF0.5/SF |
|---|---|---|---|---|---|---|---|---|
| GWP/CS | 11.65 | 11.11 | 10.85 | 11.35 | 9.13 | 8.64 | 8.34 | 8.79 |
| GWP/STS | 178.20 | 164.81 | 149.40 | 146.79 | 150.98 | 131.25 | 121.97 | 118.52 |
| GWP/FS | 124.37 | 111.81 | 105.94 | 104.94 | 107.13 | 93.91 | 89.79 | 88.15 |

**Figure 20.** GWP per unit strength of mixtures.

## 4. Conclusions

Based on the results and discussion presented in this study, the following significant conclusions can be drawn regarding the impact of JF addition on various mechanical and durability properties of concrete, with and without SF:

- The inclusion of JF is harmful to the workability, and it increased the air content of the fresh concrete. The use of SF can be helpful in improving the fresh density and air content of fresh JFRC mixtures.
- The addition of 0.3% JF resulted in an overall improvement in compressive strength of 5–10%. The combination of 10% SF and 0.3% JF led to a remarkable improvement of 26.5% and 30.6%, respectively, compared to the control sample.
- The addition of SF in the binder further improved the STS gain from adding fibre. When a 0.3% JF volume and 10% SF were added, the STS increased by 32% and 37%, respectively, compared to the control sample.
- The increase in JF content improved the overall flexural response of the concrete. The flexural ductility and post-peak deflection behaviour were improved with the increase in the JF content from 0.3% to 0.5%. JFRCs with high fibre content also showed notable yielding before experiencing the strain-softening response.
- The best improvements in both STS and FS were achieved with a 0.5% JF dosage, but the difference in the net improvement between 0.3% JF and 0.5% JF was not significant. Therefore, it can be concluded that a 0.3% JF dosage is optimal in terms of both mechanical and economic performance.
- At both 28 and 91 days, the addition of 0.5% JF and 10% SF increased the FS of the concrete by 27% and 32%, respectively. The inclusion of 0.5% JF improved the FT of the concrete by 6.4-times and 7.3-times with 0% and 10% SF addition, respectively.
- Increasing the volume of JF from 0.15% to 0.5% resulted in a 16.5% increase in the water absorption (WA) capacity of the concrete. The addition of SF helped JFRCs achieve lower WA values compared to the control concrete.
- JFRCs showed a higher level of RCIP compared to the control concrete, regardless of the fibre content. Nonetheless, the incorporation of SF helped to decrease the chloride permeability in JFRCs. This is likely due to the additional growth of C-S-H and micro-filler action, which intercept the movement of chloride ions within the microstructure of JFRCs.
- The UPV findings indicated that the density of JFRCs is lower than that of traditional concrete, which could be attributed to the presence of light-density and porous fibres. Nonetheless, incorporating SF can enhance the quality of JFRCs and result in a superior final product.
- The study found that the risk of corrosion in JFRCs increased with higher volumes of JF, and incorporating 0.5% JF resulted in a 40% reduction in electrical resistivity (ER). However, adding SF significantly improved the ER of JFRCs.
- The use of JF is slightly useful in reducing the volume-related emissions of concrete. However, it can cause significant reductions in STS and FS-related emissions. Up to a 30% net reduction in the STS-related emissions was observed due to the addition of 0.5% JF and 10% SF.

Based on the extensive experimental campaign, the optimum dosage of JF can be 0.3%, which provides the maximum mechanical strength (peak CS, STS, and FS). However, more ductility, toughness and post-peak crack resistance in flexural loading can be attained by using a high JF volume. The durability was compromised with the increasing JF content; however, conjunctive use of JF with mineral admixtures such as SF is recommended to avoid probable durability losses. It is recommended to investigate the pull-out bond behaviour of JF in cement pastes with various water-cement ratios (w/c) and mineral admixtures. The elevated temperature performance and autogenous and drying shrinkage behaviour of JFRCs still need to be examined.

Furthermore, JF is susceptible to volumetric changes, due to the absorption of moisture; therefore, the drying-wetting behaviour of concrete with JF reinforcement must be

understood and investigated in future research for applications where concrete is exposed to moisture or outdoor conditions.

**Funding:** This research received no external funding.

**Institutional Review Board Statement:** Not applicable.

**Informed Consent Statement:** Not applicable.

**Data Availability Statement:** All data shown in the manuscript. In that case, feel free to write whatever statement necessary.

**Conflicts of Interest:** The author declares no conflict of interest.

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
