# Peer review of "Effect of Silica Fume on Engineering Performance and Life Cycle Impact of Jute-Fibre-Reinforced Concrete"

_sustainability, doi:10.3390/su15118465_

Round 1

Reviewer 1 Report

An interesting topic Effect of silica fume incorporation on properties of high-performance jute fiber reinforced concrete. The work done is systematic and well presented. However, still some minor changes are required as follows;

*“Finally, the slump value of fresh concrete was checked (line 156)”: Please add Slump test results and utilized standard during the test procedures. It is of great importance to emphasize the fresh state properties, especially in the use of new types of fibers in concrete production.

*Please mention the mixer type utilized and its rpm for preparing the mixtures (high rpm can cause the breaking of the fibers, please explain).

*Please discuss the effect of aspect ratio of the  fibers on the test results.

*Please add some photos of experimental works, specimens, tests etc., if applicable.

*”Fig. 9 shows the correlations developed between different strength parameters (STS 286 vs CS, FS vs CS, and FS vs STS). The correlations for plain concrete (PCs) and jute fiber 287 reinforced concretes (JFRCs) were developed separately. It can be noticed that for PCs, CS 288 can be used to fairly predict the tensile strength parameters i.e., STS and FS” : Please verify your results with a  reliability analysis or rewrite the results. A single correlation result is too weak to support a accurate prediction system.

* What are you are your future recommendation regarding this work. Please add at the end of the conclusion

Author Response

Dear reviewer

In the attached file, I have responded to all your comments carefully.

Thank you very much for your valuable comments.

Reviewer 2 Report

The authors have submitted a well prepared paper on the interesting topic of the Effect of silica fume incorporation on properties of high-performance jute fiber reinforced concrete. The paper is clearly presented and provides interesting results. This study is valuable for the practical engineering. However, the following comments are provided to assist the authors to improve the paper:

1) The article's purpose should be clarified in detail, why this study could be beneficial, and a more in-depth conclusion in applications should be provided.

2) The use of jute fiber as a concrete mix has been extensively researched. Thus, the obtained results in the study should be compared with the existing literature.

3) Conclusions: the author should further explain this research's construction application limitations. Please describe in conclusion.

4) Please propose some improvements and directions for future research.

Thank you for considering my opinion. I encourage the authors to keep on working to improve the revised manuscript.

Minor editing of English language required

Author Response

Dear reviewer

In the attached file, I have responded to all your comments carefully.

Thank you very much for your valuable comments

Reviewer 3 Report

The content of the paper corresponds to the topic stated in the title.

The article has been divided into four chapters.

The aim of the study is to evaluate the mechanical parameters, water absorption, rapid chloride ion permeability, electrical resistivity, and ultrasonic pulse velocity of high-performance jute fiber reinforced concrete.

The aim of this study has been achieved.

The sources contain 56 references.

I recommend the article for publishing after taking into account the following remarks:

2.       Please consider correcting the naming of all chapters. Currently, the division into chapters is not fully clear.

3.       Line 100:  instead of “mixes” I would propose for example “mixtures”

4.       Instead of “Fig. 2: Gradation of binders” I would propose for example “The particle size distribution of a binder”

5.       Instead of “Fig. 3: Gradation of aggregates”  I would propose for example “The particle size distribution of an aggregate”

6.       Line 139 “2.2 Mix design and preparation preparation” - please correct.

7.       Line 173 “2.3.2 Permeability-related properties” - please correct (what properties?).

8.       Line 301 “3.2 Permeability-related properties” – please correct (the same title twice).

9.       Why the mechanical properties, i.e. compressive, flexural and tensile strength of the mixtures after 28 and 91 days, were tested?

10.   What is the w/c ratio?

Language proofreading of the article would be needed.

Author Response

(The authors gave the same response as above.)

Reviewer 4 Report

Language is clear and understandable, apart of two tiny remarks presented in the PDF file.

Author Response

(The authors gave the same response as above.)
